# Elastic Diffusion Transformer

**Jiangshan Wang** [1 2 3]  **Zeqiang Lai** [1 2]  **Jiarui Chen** [2 4]  **Jiayi Guo** [3]  **Hang Guo** [3]
**Xiu Li** [3]  **Xiangyu Yue** [1]  **Chunchao Guo** [2]

## Abstract

Diffusion Transformers (DiT) have demonstrated remarkable generative capabilities but remain highly computationally expensive. Previous acceleration methods, such as pruning and distillation, typically rely on a fixed computational capacity, leading to insufficient acceleration and degraded generation quality. To address this limitation, we propose **Elastic Diffusion Transformer (E-DiT)**, an adaptive acceleration framework for DiT that effectively improves efficiency while maintaining generation quality. Specifically, we observe that the generative process of DiT exhibits substantial sparsity (i.e., some computations can be skipped with minimal impact on quality), and this sparsity varies significantly across samples. Motivated by this observation, E-DiT equips each DiT block with a lightweight router that dynamically identifies sample-dependent sparsity from the input latent. Each router adaptively determines whether the corresponding block can be skipped. If the block is not skipped, the router then predicts the optimal MLP width reduction ratio within the block. During inference, we further introduce a block-level feature caching mechanism that leverages router predictions to eliminate redundant computations in a training-free manner. Extensive experiments across 2D image (Qwen-Image and FLUX) and 3D asset (Hunyuan3D-3.0) demonstrate the effectiveness of E-DiT, achieving up to ~2× speedup with negligible loss in generation quality. Code will be available at https://github.com/wangjiangshan0725/Elastic-DiT.

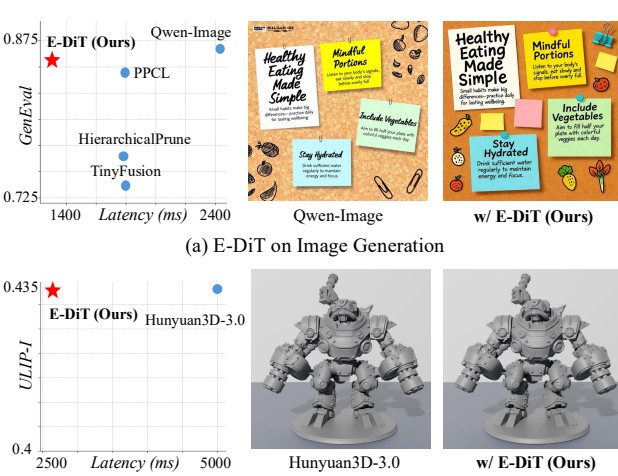

(a) E-DiT on Image Generation

(b) E-DiT on 3D Asset Generation

*Figure 1.* **Performance of Elastic Diffusion Transformer (E-DiT)** across diverse generation foundation models and modalities.

## 1. Introduction

Diffusion models have achieved remarkable progress in recent years, demonstrating strong performance across diverse modalities, including images (Labs, 2024; Wu et al., 2025; Esser et al., 2024), videos (Yang et al., 2024; Wang et al., 2025), and 3D assets (Lai et al., 2025a;d; Zhao et al., 2025). Despite these successes, they usually suffer from substantial computational overhead due to the large model sizes, which significantly limit their practical deployment. As a result, improving the efficiency of diffusion models while maintaining high generation quality has become a critical and challenging research problem.

A common strategy for accelerating diffusion models is to reduce the computational cost through pruning or distillation (Daniel Verdú, 2024; Ma et al., 2025; Kwon et al., 2025). These methods typically adopt a static design, where a fixed, smaller model architecture is uniformly applied across all denoising steps and input conditions. However, such static strategies overlook the fact that different modules within the generation process contribute unequally to the final output (Wimbauer et al., 2024; Liu et al., 2025; Zhao et al., 2024), resulting in a suboptimal trade-off between efficiency and generation quality. To mitigate this issue, several recent works explore dynamic network structures for accelerating

[1]MMLab, CUHK [2]Tencent Hunyuan [3]Tsinghua University [4]HITSZ. Correspondence to: Zeqiang Lai <laizeqiang@outlook.com>, Xiangyu Yue <xyyue@ie.cuhk.edu.hk>, Chunchao Guo <chunchaoguo@tencent.com>.

*Proceedings of the 43rd International Conference on Machine Learning*, Seoul, South Korea. PMLR 306, 2026. Copyright 2026 by the author(s).

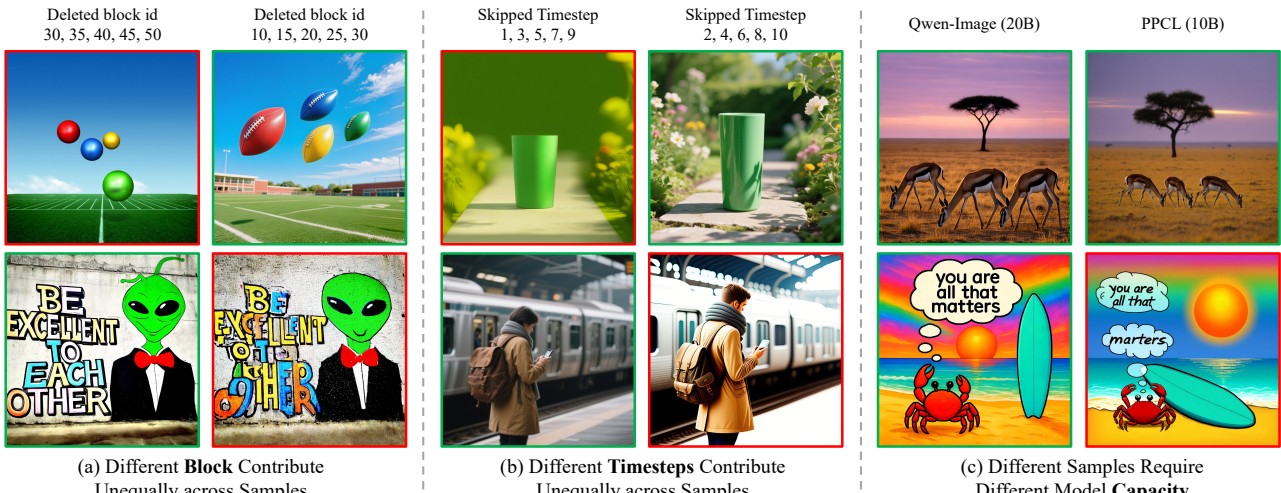

*Figure 2.* **Sample-dependent sparsity in the generation process.** We use Qwen-Image (Wu et al., 2025) to illustrate our observations.(a) Images generated after removing different subsets of DiT blocks from Qwen-Image, showing that block importance varies across samples. (b) Results obtained by skipping selected denoising timesteps using a timestep-wise feature caching strategy (Liu et al., 2025), demonstrating content-dependent sensitivity to timestep removal. (c) Comparison between images generated by the Qwen-Image base model (20B) and a pruned variant (10B) (Ma et al., 2025), highlighting that computational requirements vary with sample difficulty.

DiT models. Nevertheless, they often suffer from limited flexibility, such as requiring non-trivial architectural modifications when adapting to different backbones (Zhao et al., 2024) or activating a fixed number of parameters regardless of input complexity (Zheng et al., 2025b).

In this work, we aim to develop a general acceleration framework for diverse DiT backbones that can adaptively allocate computation according to the generated content. Specifically, we observe that the generation process exhibits significant sparsity: certain computations during denoising contribute only marginally to the final generation quality. Importantly, this sparsity is content-dependent rather than uniform across samples, as evidenced by three key aspects (Figure 2). First, different DiT blocks contribute unequally across samples (Figure 2a). Skipping a particular subset of blocks may have negligible impacts on the generation quality of some samples while severely degrading others, and different block subsets affect different samples. Second, denoising timesteps also exhibit uneven importance across samples (Figure 2b). The effect of skipping timesteps varies with the input content, similar to the behavior observed when skipping blocks. Third, computational demand correlates with the complexity of the generated samples (Figure 2c). While a lightweight model is enough to generate high-quality results for relatively easy samples, more complex samples require additional computation to maintain generation fidelity.

To exploit the sample-dependent sparsity in the diffusion generation process, we propose **Elastic Diffusion Transformer (E-DiT)**, a general and adaptive acceleration framework for diffusion transformers. E-DiT accelerates gener-

ation in a sample-adaptive manner through three complementary components: (1) adaptive block skipping, which dynamically skips entire transformer blocks whose contributions to generation are predicted to be marginal; (2) adaptive MLP width reduction, which adjusts the activated MLP width within non-skipped blocks according to sample complexity; and (3) block-wise caching, which further eliminates redundant computation by reusing intermediate features across adjacent denoising steps in a training-free manner. Concretely, each transformer block in E-DiT is equipped with a lightweight router conditioned on the input latent and the denoising timestep. The router predicts whether the block can be skipped, and for blocks that remain active, it further determines the effective MLP width within the block. During training, we jointly optimize a performance loss to preserve generation quality and an efficiency loss to encourage efficient routing decisions. Notably, we observe that the learned router predictions naturally capture the relative importance of different blocks. Leveraging this property, we further introduce a block-wise caching mechanism that uses router predictions as a criterion for feature reusing across denoising steps, enabling additional inference acceleration without extra training.

We evaluate E-DiT across multiple modalities, including Qwen-Image (Wu et al., 2025) and FLUX (Labs, 2024) for image generation, as well as Hunyuan3D-3.0 (Team, 2025) for 3D asset generation. Experimental results demonstrate that E-DiT substantially reduces inference cost with negligible degradation in quality, while being broadly applicable and compatible with various DiT backbones.

## 2. Related Work

### 2.1. Diffusion Model

Diffusion models (Ho et al., 2020; Song et al., 2020a;b; Rombach et al., 2022; Liu et al., 2022) have emerged as a prominent paradigm for high-fidelity generation, which generate the sample through denoising from a standard Gaussian noise. In recent years, diffusion transformers (DiTs) (Peebles & Xie, 2023) have shown strong scalability and have become the mainstream architecture of modern large-scale foundation models across modalities, represented by FLUX (Labs, 2024) and Qwen-Image (Wu et al., 2025) for the image generation; HunyuanVideo (Kong et al., 2024), CogVideoX (Yang et al., 2024), and Wan2.1 (Wang et al., 2025) for video generation; Hunyuan3D (Lai et al., 2025a; Zhao et al., 2025), LATTICE (Lai et al., 2025b), and Trellis (Xiang et al., 2024) for 3D asset generation.

### 2.2. Diffusion Model Acceleration

High-quality generation usually requires dozens of denoising steps and expensive transformer computations at each step, resulting in substantial inference latency and compute cost. To mitigate this issue, prior works have explored acceleration via step distillation (Cheng et al., 2025; Lu & Song, 2024; Zheng et al., 2025a; Geng et al., 2025a; Lu et al., 2022; Song et al., 2023; Lai et al., 2025c), model architecture compression (Ma et al., 2025; Kwon et al., 2025; Daniel Verdú, 2024; Fang et al., 2025), and training-free methods such as sparse attention (Zhang et al., 2025; Xi et al., 2025), token pruning (Bolya et al., 2022; Wang et al., 2024a; Fang et al., 2026) and feature caching (Selvaraju et al., 2024; Wimbauer et al., 2024; Kahatapitiya et al., 2024; Guo et al., 2026). However, these methods usually adopt fixed network structures and parameters for all samples, ignoring sample-specific variability, which could lead to a less favorable quality-efficiency trade-off.

### 2.3. Dynamic Neural Networks

Dynamic neural networks (Han et al., 2021) adapt computation to individual inputs by conditionally activating different parts of the network to reduce redundancy. Representative works include conditional computation with dynamic depth or width, as well as token- or head-level sparsification in Transformers (Meng et al., 2022; Song et al., 2021; Wang et al., 2024c; Rao et al., 2021; Liang et al., 2022; Li et al., 2021). Recent efforts have recently explored dynamic mechanisms in diffusion transformers, including converting dense backbones into Mixture-of-Experts structures (Zheng et al., 2025b; Cheng et al.; Wei et al., 2025; Shi et al., 2025) and dynamically selecting attention heads and tokens (Zhao et al., 2024). However, these methods often exhibit limited flexibility, such as fixed expert grouping or activation patterns, and require non-trivial structural redesign when applied to different backbones or modalities.

## 3. Method

In this section, we present the design of Elastic Diffusion Transformer (E-DiT) in detail. We first introduce the model architecture of E-DiT, including the router design and adaptive mechanisms for block skipping and MLP width reduction. Next, we describe the training strategy based on a joint quality–efficiency objective. Finally, we present the inference process of E-DiT, where block-wise caching is employed to further accelerate generation by exploiting temporal redundancy across denoising steps.

### 3.1. Preliminaries

**Rectified Flow** (Liu et al., 2022) formulates generative modeling as learning a linear transport path between the data distribution $\pi_0$ and a noise distribution $\pi_1$ via an ordinary differential equation (ODE):

$$d\mathbf{x}_t = v(\mathbf{x}_t, t)\, dt, \quad t \in [0, 1], \tag{1}$$

where the velocity field $v$ is parameterized by a network $\epsilon_\theta$. Given $\mathbf{x}_0 \sim \pi_0$ and $\mathbf{x}_1 \sim \pi_1$, the trajectory is defined as $\mathbf{x}_t = (1 - t)\mathbf{x}_0 + t\mathbf{x}_1$, yielding the training objective

$$\min_\theta \ \mathbb{E}_{t \sim \mathcal{U}(0,1)} \left[ \|(\mathbf{x}_1 - \mathbf{x}_0) - \epsilon_\theta(\mathbf{x}_t, t)\|^2 \right]. \tag{2}$$

During inference, samples are generated by numerically solving the learned ODE from Gaussian noise (Lu et al., 2022; Wang et al., 2024b), using a solver such as Euler. Compared to DDPM (Ho et al., 2020), Rectified Flow achieves high-quality generation with substantially fewer sampling steps, making it particularly suitable for large-scale generative models (Wu et al., 2025; Labs, 2024; Lai et al., 2025a).

**Multi-Modal Diffusion Transformer** (DiT) (Peebles & Xie, 2023) demonstrates the scalability and effectiveness of Transformer architectures for diffusion-based generative modeling. Extending this framework, MMDiT (Labs, 2024; Wu et al., 2025) integrates conditioning information via self-attention applied jointly to both data and condition tokens. An MMDiT model comprises a stack of Transformer blocks, each consisting of a joint multi-head self-attention (MHSA) module over data and condition tokens, followed by a multi-layer perceptron (MLP). The MLP consists of two linear layers with an intermediate non-linear activation. Specifically, given the input $\mathbf{z} \in \mathbb{R}^{L \times D}$, where $L$ is the sequence length and $D$ is the feature dimension, the standard MLP first projects $\mathbf{z}$ to a higher-dimension $H$, applies a non-linear activation (e.g., GELU), and then projects it back to the original dimension $D$, i.e.,

$$\text{MLP}(\mathbf{z}) = \sigma(\mathbf{z}\mathbf{W}_1)\mathbf{W}_2, \tag{3}$$

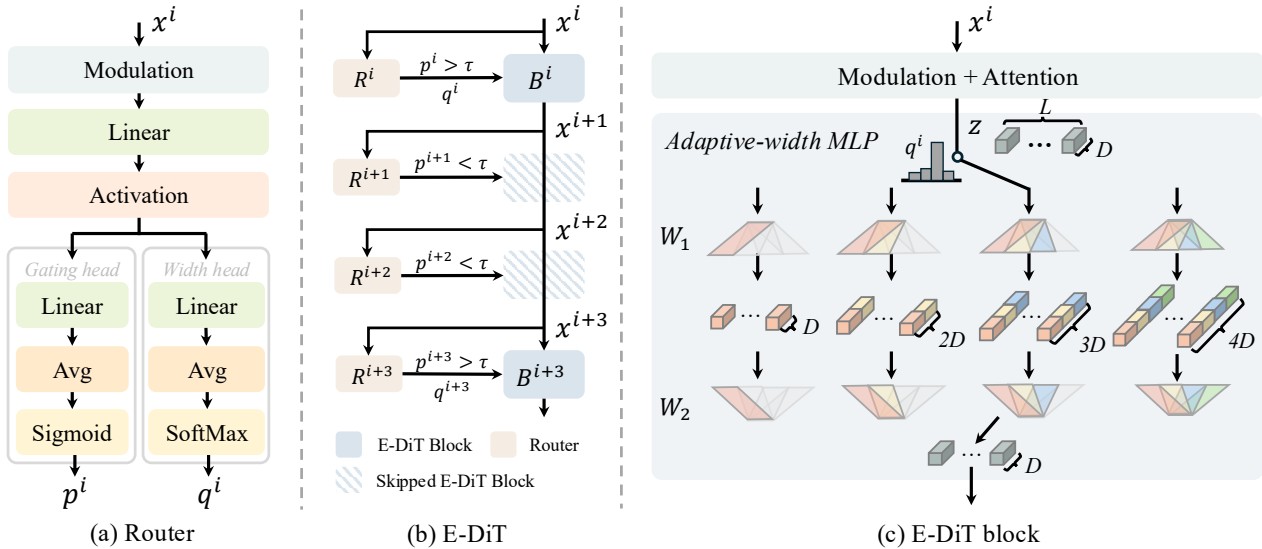

*Figure 3.* **Overall pipeline of Elastic Diffusion Transformer (E-DiT).** (a). The architecture of the router, which predicts $p_g$ and $p_w$, indicating whether the block can be skipped and the width of the MLP within the block, respectively. (b). The overall structure of the E-DiT, where each transformer block is equipped with a router. (c). The structure of the transformer block within the E-DiT, where the width of the MLP is adaptively reduced according to the router's prediction.

where $\mathbf{W}_1 \in \mathbb{R}^{D \times H}$ and $\mathbf{W}_2 \in \mathbb{R}^{H \times D}$ are linear projections and $\sigma(\cdot)$ is the activation function. We refer to the ratio $H/D$ as the width of the MLP, which is typically set to 4 in most generation models.

### 3.2. Model Designs

**Router Architecture.** Given a Diffusion Transformer (DiT) with $n$ blocks $\{\mathbf{B}^i\}_{i=1}^n$, we equip each block $\mathbf{B}^i$ with a lightweight router $\mathbf{R}^i$ to enable adaptive computation. For a given input, $\mathbf{R}^i$ first predicts whether the block should be activated; if activated, it further predicts the appropriate MLP width within that block.

Formally, let $\mathbf{x}_t^i \in \mathbb{R}^{L \times D}$ denote the input latent to block $\mathbf{B}^i$ at diffusion step $t$. Within the router, we first apply timestep-conditioned modulation using Layer Normalization (LN) followed by element-wise scaling and shifting:

$$\widetilde{\mathbf{x}}_t^i = \left(1 + \boldsymbol{\gamma}(t)\right) \odot \mathrm{LN}(\mathbf{x}_t^i) + \boldsymbol{\delta}(t), \qquad (4)$$

where $\boldsymbol{\gamma}(t), \boldsymbol{\delta}(t) \in \mathbb{R}^D$ are timestep-dependent scale and shift parameters obtained via a linear projection of the timestep embedding $\mathbf{E}(t) \in \mathbb{R}^D$, and $\odot$ denotes element-wise multiplication. The modulated features $\widetilde{\mathbf{x}}_t^i$ are then projected and passed through a non-linear activation:

$$\mathbf{h} = \sigma\left(\widetilde{\mathbf{x}}_t^i \mathbf{W}\right) \in \mathbb{R}^{L \times H_r}. \qquad (5)$$

where $\mathbf{W} \in \mathbb{R}^{D \times H_r}$ and $H_r \ll D$ to keep the router lightweight and efficient.

Based on $\mathbf{h}$, the router produces two outputs via separate linear heads and global averaging, i.e., (1). A gating logit $\ell_t^i$ for adaptive block skipping. (2) A width logit vector $\mathbf{u}_t^i \in \mathbb{R}^4$ for adaptive MLP width reduction.

$$\ell_t^i = \frac{1}{L} \sum_{j=1}^{L} \mathbf{h}[j,:]\mathbf{W}_g, \qquad \mathbf{u}_t^i = \frac{1}{L} \sum_{j=1}^{L} \mathbf{h}[j,:]\mathbf{W}_w, \quad (6)$$

where $\mathbf{W}_g \in \mathbb{R}^{H_r \times 1}$ and $\mathbf{W}_w \in \mathbb{R}^{H_r \times 4}$ denote the parameters of the gating and width heads, respectively.

**Adaptive Block Skipping.** Given the scalar logit $\ell_t^i \in \mathbb{R}$ predicted by the router, we convert it to a probability via the sigmoid function: $p_t^i = \sigma(\ell_t^i) \in [0, 1]$. The corresponding block $\mathbf{B}^i$ is skipped if $p_t^i$ falls below a predefined threshold $\tau$ (set to 0.5 in our experiments), allowing the model to eliminate redundant computation.

During training, the discrete block-skipping operation is non-differentiable. To address this, we adopt the Straight-Through Estimator (STE) (Bengio et al., 2013), which allows gradients to propagate through the gating decisions. Specifically, we define the gate variable as:

$$g_t^i = \mathbb{1}[p_t^i \geq \tau] + p_t^i - \mathrm{StopGrad}(p_t^i), \qquad (7)$$

where $\mathbb{1}[\cdot]$ is the indicator function, returning 1 if the input is true and 0 otherwise. The output for the $i$th block is then computed as:

$$\mathbf{x}_t^{i+1} = \mathbf{x}_t^i + g_t^i \cdot \left(\mathbf{B}^i(\mathbf{x}_t^i) - \mathbf{x}_t^i\right), \qquad (8)$$

To encourage computational efficiency, we regularize the routing by constraining the average gate probability $\bar{p} = \frac{1}{n} \sum_{i=1}^n p_t^i$ to match a target $\rho_g \in (0, 1)$ via the gating loss:

$$\mathcal{L}_{\mathrm{gating}} = (\bar{p} - \rho_g)^2. \qquad (9)$$

During inference, we directly skipped the blocks with $p_t^i < \tau$ to achieve acceleration, i.e.,

$$\mathbf{x}_t^{i+1} = \begin{cases} \mathbf{x}_t^i, & p_t^i < \tau, \\ \mathbf{B}^i(\mathbf{x}_t^i), & p_t^i \geq \tau. \end{cases} \quad (10)$$

**Adaptive MLP Width Reduction.** For blocks that are not skipped, we further reduce computation by dynamically adjusting the MLP width from the original $H/D = 4$ according to a set of predefined reduction ratios $\mathcal{S} = \{\frac{1}{4}, \frac{1}{2}, \frac{3}{4}, 1\}$. Specifically, given the router prediction $\mathbf{u}_t^i \in \mathbb{R}^4$, we first compute width probabilities via softmax:

$$\mathbf{q}_t^i = \mathrm{softmax}(\mathbf{u}_t^i) \in \mathbb{R}^4. \quad (11)$$

The width with the highest probability is then selected:

$$k = \mathrm{argmax}_j \mathbf{q}_t^i[j], \quad \hat{s}_t^i = \mathcal{S}[k]. \quad (12)$$

During training, we implement the adaptive MLP by masking intermediate activations to preserve differentiability:

$$\mathrm{MLP}_{\mathrm{adapt}}(\mathbf{z}) = \left(\sigma(\mathbf{z}\mathbf{W}_1) \odot \mathbf{m}(\hat{s}_t^i)\right)\mathbf{W}_2, \quad (13)$$

where $\odot$ denotes element-wise multiplication and $\mathbf{m}(\hat{s}_t^i) \in \{1, 0\}^H$ represents a mask which only keeps the first $\hat{s}_t^i \cdot H$ part of the feature along the hidden dimension.

To encourage more efficient width selection, we regularize the averaged MLP width across all non-skipped blocks. Formally, the average width reduction for the block $\mathbf{B}^i$ at the timestep $t$ is defined as $r_t^i = \sum_{j=1}^4 \mathbf{q}_t^i[j]\, s[j]$. Since width allocation is only meaningful when the block is not skipped, we mask out skipped blocks through $\mathbb{1}[p_t^i \geq \tau]$. The masked global average width reduction is computed as

$$\bar{r} = \frac{\sum_{i=1}^n \mathbb{1}[p_t^i \geq \tau]\, r_t^i}{\sum_{i=1}^n \mathbb{1}[p_t^i \geq \tau]}. \quad (14)$$

We encourage $\bar{r}$ to match a target width budget $\rho_w \in (0, 1)$ via $\mathcal{L}_{\mathrm{width}}$:

$$\mathcal{L}_{\mathrm{width}} = \left(\bar{r} - \rho_w\right)^2. \quad (15)$$

During inference, the adaptive MLP width is implemented by explicit matrix slicing to avoid computation on deactivated channels:

$$\mathrm{MLP}_{\hat{s}_t^i}(\mathbf{z}) = \sigma(\mathbf{z}\widetilde{\mathbf{W}}_1)\widetilde{\mathbf{W}}_2, \quad (16)$$

where $\widetilde{\mathbf{W}}_1 = \mathbf{W}_1[:, : H \cdot \hat{s}_t^i]$, $\widetilde{\mathbf{W}}_2 = \mathbf{W}_2[: H \cdot \hat{s}_t^i, :]$, yielding actual acceleration by skipping computation on the deactivated channels.

### 3.3. Training and Inference

**Training Pipeline.** We train E-DiT end-to-end on top of a pretrained diffusion Transformer. At the start of training, all routers are set to be fully open, i.e., each block is activated with full MLP width, ensuring that training begins from the original dense model behavior and avoiding unstable

---

**Algorithm 1** Pseudo-Code for E-DiT Inference

**Input:**
- $\mathbf{x}_T \sim \mathcal{N}(\mathbf{0}, \mathbf{I})$      *Initial Gaussian Noise*
- $T$      *Number of denoising steps*
- $\{\mathbf{B}^i\}_{i=1}^N$      *Network blocks*
- $\{\mathbf{R}^i\}_{i=1}^N$      *Routers*
- $\{\mathcal{C}^i \leftarrow \emptyset\}_{i=1}^N$      *Feature bank*
- $\tau$      *Block skipping threshold*
- $\delta$      *Borderline margin*
- $K$      *Maximum reuse limit*

**Denoising Process:**
**for** $t = T, T-1, \ldots, 0$ **do**
  $\mathbf{x}_t^1 \leftarrow \mathbf{x}_t$
  **for** $i = 1, 2, \ldots, N$ **do**
    $p_t^i, \mathbf{q}_t^i \leftarrow \mathbf{R}^i(\mathbf{x}_t^i, t)$
    **if** $p_t^i < \tau$ **then**
      $\mathbf{x}_t^{i+1} \leftarrow \mathbf{x}_t^i$      *Directly skip the block*
      **continue**
    **end if**
    **if** $\tau \leq p_t^i \leq \tau + \delta$ **and** $\mathcal{C}^i \neq \emptyset$ **and** $k^i < K$ **then**
      $\mathbf{x}_t^{i+1} \leftarrow \mathbf{x}_t^i + \Delta^i$ *Skip the block through feature reusing*
      $k^i \leftarrow k^i + 1$
      **continue**
    **end if**
    $\mathbf{x}_t^{i+1} \leftarrow \mathbf{B}^i(\mathbf{x}_t^i, \mathbf{q}_t^i)$   *Inference with adaptive MLP width*
    $\Delta^i \leftarrow \mathbf{x}_t^{i+1} - \mathbf{x}_t^i$      *Cache the residual*
    $k^i \leftarrow 0$      *Reset reuse counter*
    $\mathcal{C}^i \leftarrow (\Delta^i, k^i)$
  **end for**
  $\mathbf{x}_{t-1} \leftarrow \mathrm{DENOISESTEP}(\mathbf{x}_t^{N+1}, t)$      *Update latent*
**end for**
**Output:** $\mathbf{x}_0$      *Final denoised sample*

---

early-stage optimization. During training, given a mini-batch of latent inputs and randomly sampled timesteps, each router predicts the block gate probability $p_t^i$ and the width distribution $\mathbf{q}_t^i$ for each block $\mathbf{B}_i$ (Sec. 3.2). The overall training objective combines quality and efficiency:

$$\mathcal{L} = \mathcal{L}_{\mathrm{perf}} + \lambda\, \mathcal{L}_{\mathrm{eff}}, \quad (17)$$

where $\lambda$ balances the two terms (we set $\lambda = 1$ in experiments). The performance loss $\mathcal{L}_{\mathrm{perf}}$ is the flow-matching objective in Equation (2), used by the underlying diffusion backbone, while the efficiency regularization $\mathcal{L}_{\mathrm{eff}} = \mathcal{L}_{\mathrm{gating}} + \mathcal{L}_{\mathrm{width}}$ encourages sample- and timestep-adaptive routing and width allocation. This formulation allows E-DiT to learn dynamic, content-dependent computation while preserving the generation quality of the original dense model.

**Inference Pipeline & Block-wise Caching.** During inference, E-DiT dynamically adapts both block execution and MLP widths. For each block $\mathbf{B}^i$, the router predicts a gating probability $p_t^i \in [0, 1]$ and a width distribution $\mathbf{q}_t^i \in \mathbb{R}^4$ (Sec. 3.2) at each denoising step $t$. A block is skipped when $p_t^i < \tau$ (we set $\tau = 0.5$); otherwise, it is activated with the selected MLP width.

While adaptive block skipping and MLP width reduction al-

ready eliminate most redundant computation with minimal quality loss, we observe that some active blocks ($p_t^i \geq \tau$) have gating probabilities close to the threshold, suggesting further potential for acceleration. To exploit this, we define a borderline region $p_t^i \in [\tau, \tau + \delta]$, where blocks are not directly skipped but likely contribute marginally. For such blocks, we leverage temporal redundancy across denoising steps via a block-wise caching mechanism, reusing intermediate features to further reduce computation.

Specifically, at timestep $t$, when the $\mathbf{B}^i$ is activated, we compute its residual update as

$$\Delta^i = \mathbf{B}^i(\mathbf{x}_t^i, \mathbf{q}_t^i) - \mathbf{x}_t^i, \tag{18}$$

and store it in a feature bank $\mathcal{C}^i$. At a later timestep $\tilde{t}$, if the gating probability $p_{\tilde{t}}^i$ of this block falls within the borderline region $[\tau, \tau + \delta]$ and a cached residual in $\mathcal{C}^i$ is available, we skip the full block computation and update the latent via

$$\mathbf{x}_{\tilde{t}}^{i+1} = \mathbf{x}_{\tilde{t}}^i + \Delta^i. \tag{19}$$

Otherwise, a full forward pass is performed, and the feature bank is refreshed with the newly computed residual. To prevent error accumulation, each cached residual is reused at most $K$ times before recomputation.

Unlike prior caching methods that require designing complicated criteria to determine when to reuse features, E-DiT naturally leverages the router prediction $p_t^i$ as a principled cache indicator, providing a simple yet effective mechanism to further reduce redundant computation. Overall, the inference process of E-DiT is illustrated in Algorithm 1.

## 4. Experiments

### 4.1. Experimental Setup

**Implementation Details.** We implement E-DiT on several representative foundation models for generative modeling across both 2D image and 3D asset modalities, including Qwen-Image (Wu et al., 2025), FLUX (Labs, 2024), and Hunyuan3D-3.0 (Team, 2025). For Qwen-Image, we train two versions of our E-DiT, termed *E-DiT-base* and *E-DiT-turbo*. Specifically, E-DiT-base adopts $\rho_g = 0.6$ for adaptive block skipping, $\rho_w = 0.65$ for adaptive MLP width reduction, and block-wise caching with $\delta = 0.01$ and $K = 5$. E-DiT-turbo applies more aggressive acceleration, with $\rho_g = 0.5$, $\rho_w = 0.6$, $\delta = 0.015$, and $K = 10$. For FLUX, we set $\rho_g = 0.5$, $\rho_w = 0.6$, and use block-wise caching with $\delta = 0.01$ and $K = 3$. For Hunyuan3D-3.0, we use $\rho_g = 0.45$, $\rho_w = 0.5$, $\delta = 0.015$, and $K = 5$. The number of denoising steps $T$ is set to 30, 28, and 5 for Qwen-Image, FLUX, and Hunyuan3D-3.0, respectively, following their default configurations. Following previous work (Cheng et al., 2025; Chen et al., 2025d; Geng et al., 2025b), we train image generation models on the BLIP3o-60K (Chen et al., 2025c) and ShareGPT-4o (Chen et al.,

*Table 1.* Quantitative results for text-to-image generation on Qwen-Image. L. denotes inference latency (milliseconds).

| Methods | L.↓ | DPG↑ | GenEval↑ | T2I-CompBench↑ | | |
| --- | --- | --- | --- | --- | --- | --- |
| | | | | B-VQA | UniDet | S-CoT |
| Base model | 2431 | 88.9 | 0.870 | 0.709 | 0.532 | 82.47 |
| TinyFusion | 1789 | 80.7 | 0.739 | 0.689 | 0.464 | 78.99 |
| HP | 1786 | 83.3 | 0.766 | 0.706 | 0.487 | 79.94 |
| PPCL | 1792 | 87.9 | 0.847 | 0.750 | 0.524 | 82.15 |
| **E-DiT-base** | 1702 | 88.1 | 0.893 | 0.719 | 0.536 | 82.34 |
| **E-DiT-turbo** | 1283 | 85.4 | 0.853 | 0.711 | 0.519 | 81.68 |

*Table 2.* Quantitative results for text-to-image generation on FLUX.1-dev. L. denotes inference latency (milliseconds).

| Methods | L.↓ | DPG↑ | GenEval↑ | T2I-CompBench↑ | | |
| --- | --- | --- | --- | --- | --- | --- |
| | | | | B-VQA | UniDet | S-CoT |
| Base model | 715 | 83.8 | 0.665 | 0.640 | 0.426 | 78.57 |
| Dense2MoE | 513 | 76.2 | 0.475 | 0.494 | 0.340 | 77.50 |
| DyDiT | 423 | 80.3 | 0.676 | - | - | - |
| TinyFusion | 534 | 77.2 | 0.511 | 0.584 | 0.369 | 74.17 |
| HP | 543 | 75.7 | 0.503 | 0.579 | 0.371 | 74.99 |
| PPCL | 535 | 80.0 | 0.605 | 0.615 | 0.391 | 78.15 |
| **E-DiT** | 374 | 80.5 | 0.671 | 0.612 | 0.402 | 77.91 |

2025b) datasets, totaling approximately 100K images. For 3D asset generation, we use the internal dataset. All experiments of training are conducted on 32 NVIDIA H20 GPUs. Inference is conducted on a single NVIDIA H20 GPU.

**Baselines.** For image generation, we compare E-DiT against several state-of-the-art pruning-based acceleration methods, including FLUX.1 Lite (Daniel Verdú, 2024), Tiny-Fusion (Fang et al., 2025), HierarchicalPrune (HP) (Kwon et al., 2025), and PPCL (Ma et al., 2025). We additionally include dynamic acceleration methods, namely Dense2MoE (Zheng et al., 2025b) and DyDiT (Zhao et al., 2024). For PPCL and DyDiT, we directly report the results from their original papers, while results for the remaining baselines are taken from the PPCL benchmark. For visual comparison, we compare our methods with open-source baselines. The prompts for visual comparison are provided in the Appendix. For 3D asset generation, we compare E-DiT with the unaccelerated Hunyuan3D-3.0 baseline. More information about baselines is provided in the Appendix.

**Evaluation Metrics.** For image generation, we report results on DPG-Bench (Hu et al., 2024), GenEval (Ghosh et al., 2023), and T2I-CompBench (Huang et al., 2025). For 3D asset generation, performance is evaluated using ULIP (Xue et al., 2023) and Uni3D (Zhou et al., 2023) scores, together with qualitative comparisons. Inference efficiency is measured by the per-step latency on a single H20 GPU for both the baseline methods and E-DiT. We also provide visual comparisons between E-DiT and baselines to illustrate the effectiveness of our method.

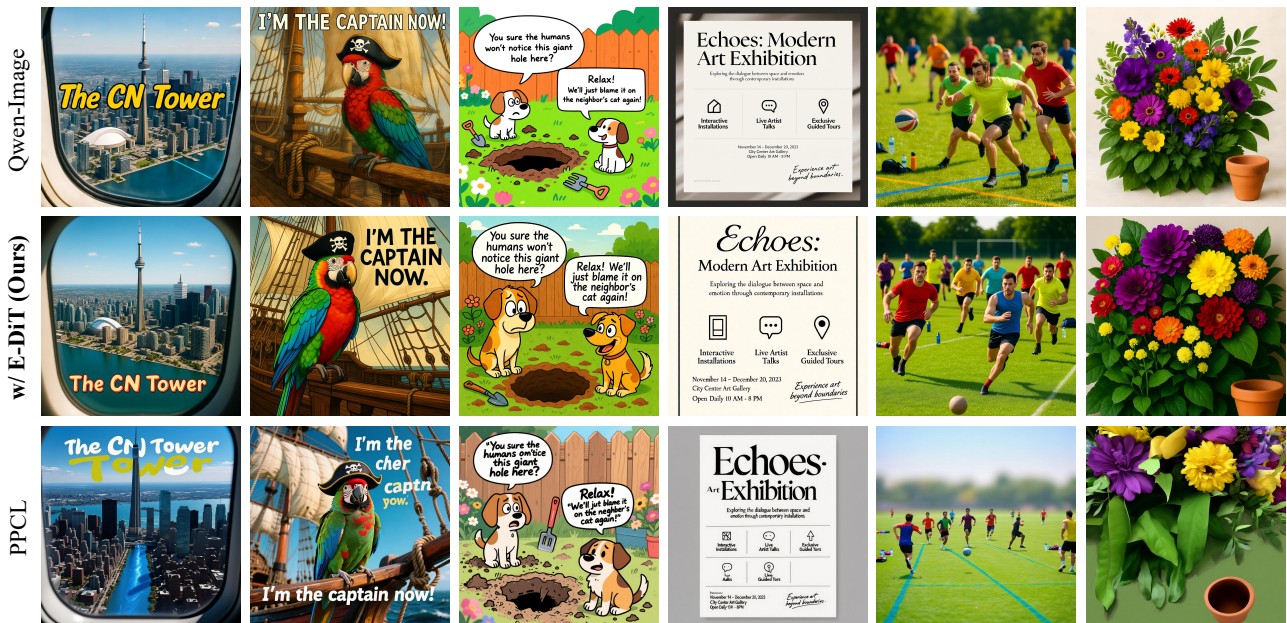

*Figure 4.* Visual comparisons between E-DiT-turbo and open-source baselines based on Qwen-Image.

*Table 3.* Quantitative results of shape generation methods.

| Methods | ULIP-I↑ | Uni3D-I↑ | Latency (ms)↓ |
|---|---|---|---|
| Hunyuan3D-3.0 | 0.1446 | 0.4334 | 5012 |
| **E-DiT** | **0.1473** | **0.4332** | **2587** |

*Table 5.* Ablation study of different initialization strategies.

| Initialization Strategy | DPG↑ | GenEval↑ |
|---|---|---|
| Random init | 78.6 | 0.801 |
| Full-capacity init | 85.4 | 0.853 |

*Table 4.* Ablation study of different acceleration components.

| Skip Block | Reduced Width | Block Cache | L.↓ | DPG↑ | GenEval↑ |
|---|---|---|---|---|---|
| *non-adaptive* | | | 1514 | 83.7 | 0.843 |
| ✓ | – | – | 1967 | 87.6 | 0.895 |
| ✓ | ✓ | – | 1643 | 85.8 | 0.857 |
| ✓ | ✓ | ✓ | 1283 | 85.4 | 0.853 |

### 4.3. Image-to-3D Generation

For 3D asset generation, we implement E-DiT on Hunyuan3D-3.0 and compare model performance before and after acceleration. For quantitative evaluation, we report ULIP-I (Xue et al., 2023) and Uni3D-I (Zhou et al., 2023) scores, which measure the similarity between the generated meshes and the input images (Table 3). The results show that E-DiT achieves nearly a 2× speedup while maintaining comparable generation quality. Visual comparisons (Figure 5) also demonstrate that the accelerated model preserves the ability to generate high-fidelity geometric details.

### 4.4. Discussions

**Ablation of E-DiT Components.** We systematically evaluate the contribution of each E-DiT component (Table 4). Both adaptive block skipping and MLP width reduction individually provide notable acceleration. Building on these, block-wise caching further enhances the efficiency, achieving the lowest latency with negligible quality degradation. To assess the importance of adaptive computing, we also train a non-adaptive baseline that applies random block removal and width reduction prior to training, calibrated to match the latency of our adaptive model. Despite comparable latency, this static model shows substantially worse

### 4.2. Text-to-Image Generation

We evaluate E-DiT on Qwen-Image and FLUX, with quantitative results respectively reported in Table 1 and Table 2. On both models, E-DiT can achieve roughly 2× speedup over the base architectures while maintaining consistent performance across benchmarks. We note that DyDiT achieves similar quality metrics on FLUX compared with E-DiT, but our method achieves lower latency. Moreover, E-DiT and DyDiT are not exclusive and could be combined to achieve further acceleration. Visual comparisons in Figure 4 further demonstrate that the accelerated models preserve the ability to synthesize complex visual content, including accurate long-text rendering and coherent spatial compositions.

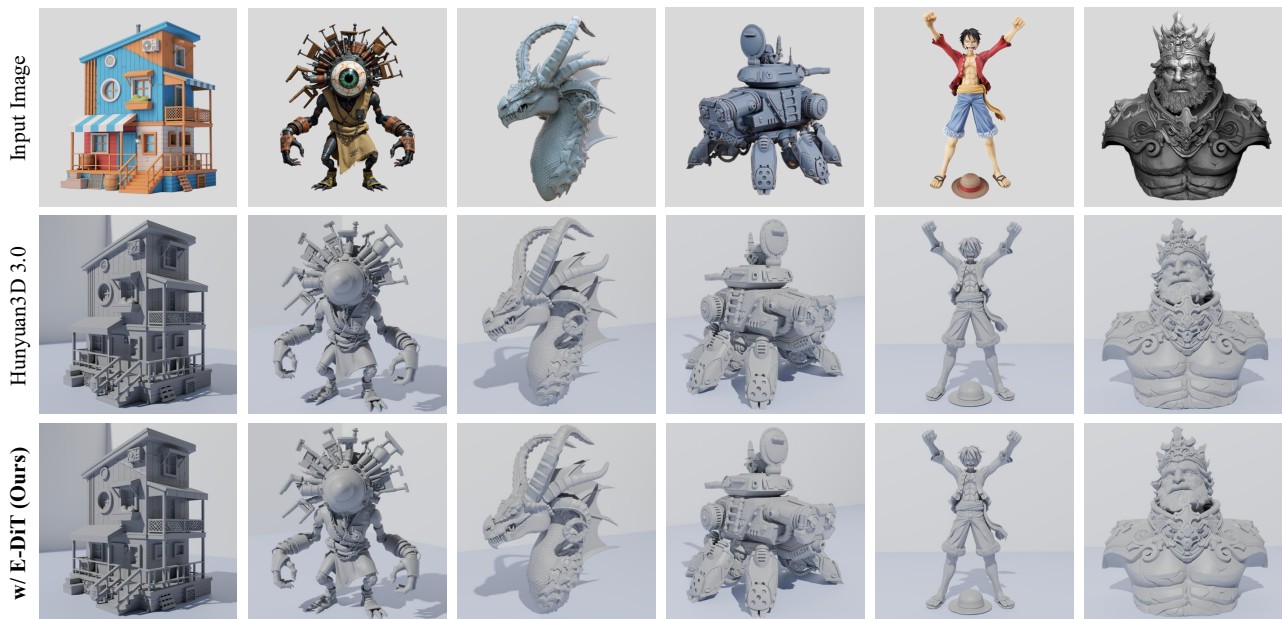

*Figure 5.* Visual comparison of Hunyuan3D 3.0 without and with E-DiT

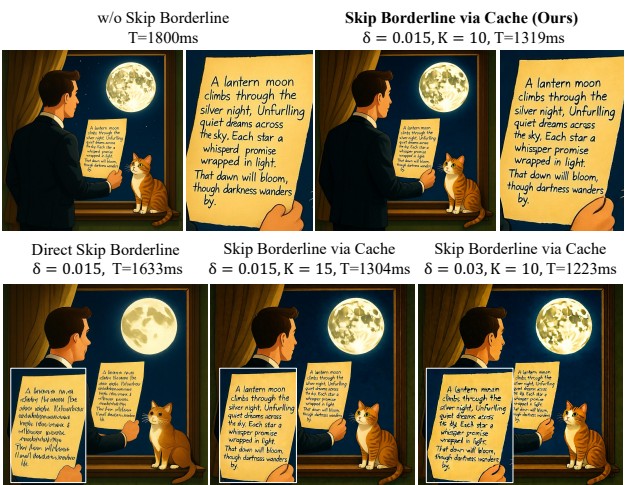

*Figure 6.* Ablation study of block-wise caching.

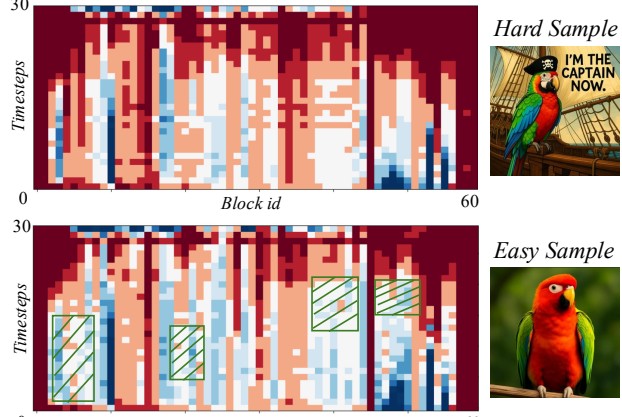

*Figure 7.* Visualization of the router predictions.

generation quality, highlighting that adaptive computing is essential for achieving an effective trade-off between efficiency and quality.

**Ablation of Block-wise Caching.** We analyze the effectiveness of block-wise caching and the impact of its hyperparameters (Figure 6). First, we observe that directly skipping those borderline blocks results in a noticeable degradation in generation quality. In addition, the acceleration achieved by direct skipping differs from that of block-wise caching, although the threshold is set to be the same (e.g., $\delta = 0.015$ in Figure 6), as the two strategies induce different latent

representations, leading to different router predictions in subsequent steps and blocks. Increasing the maximum reuse count $K$ of cached features has only a marginal effect on generation quality and shows diminishing benefits for latency reduction beyond a certain point. In contrast, the cache activation threshold $\delta$ plays a more critical role: a larger $\delta$ results in caching a greater fraction of blocks, substantially degrading generation quality.

**Ablation of Initialization.** We initialize the router to preserve the full model capacity, i.e., no blocks are skipped, and all MLPs operate at their full width at the beginning of training. We find that this initialization significantly im-

proves training stability and consistently leads to better final performance. This suggests that gradually learning adaptive acceleration from a full-capacity starting point is crucial for effective optimization.

**Analysis of Router Behavior.** We visualize router predictions across denoising timesteps in Figure 7. Red points indicate gating probabilities above 0.5, while blue points denote probabilities below 0.5; lighter red points correspond to values near the threshold, likely to be handled by block-wise caching. The visualization reveals that certain DiT blocks (e.g., the first two and the last three blocks) are consistently critical to generation quality and are rarely skipped. Furthermore, E-DiT demonstrates input-dependent behavior: for simpler inputs, such as images with clear layouts and blurred backgrounds, routers assign lower probabilities to more blocks, enabling more aggressive skipping and faster inference. In contrast, more complex inputs with intricate backgrounds or text trigger higher probabilities across more blocks, reflecting the need for increased computation to preserve generation quality.

## 5. Conclusion

In this work, we propose Elastic Diffusion Transformer (E-DiT), a general adaptive framework for efficient diffusion generation. In E-DiT, each DiT block is equipped with a router that dynamically predicts whether the block can be skipped. For non-skipped blocks, the router further determines the appropriate MLP width. During inference, we introduce a block-wise caching mechanism that leverages router predictions to reduce temporal redundancy across denoising steps. Extensive experiments on both image and 3D asset generation demonstrate the effectiveness of E-DiT, achieving roughly $2\times$ acceleration on Qwen-Image, FLUX, and Hunyuan3D-3.0 with negligible quality degradation.

## Impact Statement

This paper presents work whose goal is to advance the field of machine learning. There are many potential societal consequences of our work, none of which we feel must be specifically highlighted here.

## Acknowledgement

This work is partially supported by the National Natural Science Foundation of China (No. 62306261), HK RGC-Early Career Scheme (No. 24211525), ITSP Platform Project (No. ITS/600/24FP) and the SHIAE Grant (No. 8115074). This study was supported in part by the Centre for Perceptual and Interactive Intelligence, a CUHK-led InnoCentre under the InnoHK initiative of the Innovation and Technology Commission of the Hong Kong Special Administrative Region Government. This work is also partially supported by Hong Kong RGC Strategic Topics Grant (No. STG1/E-403/24-N), and CUHK-CUHK(SZ)-GDST Joint Collaboration Fund (No. YSP26-4760949).

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

## A. Detailed Experimental Setup

**Prompts for Image Generation.** Here, we provide the prompts used for generating images in the main paper.

Figure 1:

- A contemporary and playful slide designed like a vibrant bulletin board, displaying textual content pinned on colorful sticky notes arranged casually yet clearly across a textured corkboard background. At the top-left corner, the bold, handwritten-style heading "Healthy Eating Made Simple" stands out clearly, accompanied by a short note: "Small habits make big differences—practice daily for lasting wellbeing". Nearby toward the upper center area, a bright yellow note reads "Mindful Portions", with smaller explanatory text: "Listen to your body's signals, eat slowly and stop before overly full". Just below and to the right, a pastel-green note labeled "Include Vegetables" explains succinctly: "Aim to fill half your plate with colorful veggies each day". Moving leftward toward the lower section, a soft-blue sticky note titled "Stay Hydrated" contains the brief sentence: "Drink sufficient water regularly to maintain energy and focus". Decorative elements on the corkboard, such as small doodled fruits, vegetables, and paper clips scattered around the borders, enhance visual warmth and highlight the approachable nature of the presented messages.

Figure 2(a):

- In the open expanse of a school's sports field, under the clear blue sky of a radiant sunny day, four vibrant American footballs are captured in mid-flight. The footballs, featuring hues of red, blue, yellow, and green, are spherical in shape, contrasting sharply with the green turf below. Each ball glistens in the sunlight as they arc gracefully above the field, momentarily suspended against the backdrop of a few wispy clouds.

- A vibrant mural on a red brick wall features the inspirational phrase "BE EXCELLENT TO EACH OTHER" in bold, black lettering. Next to the text, there's a whimsical graffiti depiction of a green alien donning a sleek black tuxedo, complete with a bow tie. In the foreground, a bright yellow fire hydrant stands out on the gray concrete sidewalk, adding a pop of color to the urban scene.

Figure 2(b):

- In the midst of a vibrant garden, a cylindrical green cup stands alone on a stone path, its surface reflecting the bright afternoon sunlight. The cup, with a smooth finish, is surrounded by blossoming flowers and lush greenery. The shadows of nearby plants dance on the cup as gentle breezes sway their leaves.

- A traveler stands in a busy train station, clad in a tan jacket that falls to their mid-thigh, paired with a snugly fitted gray scarf around their neck. On their back, a worn brown backpack suggests a journey in progress. They are intently gazing at the cell phone in their hands, perhaps checking a schedule or reading a message. Before them, a sleek silver train rests on the track, its doors poised to open and welcome passengers to the next leg of their travels.

Figure 2(c):

- Three graceful antelopes are seen grazing on the sparse, golden grasses of the sprawling savannah under the soft, purpling skies of dawn. The silhouette of an acacia tree punctuates the horizon as the first light of day gently begins to illuminate the vast, open plain. With delicate movements, the animals move slowly, their tan and white coats blending subtly with the earthy tones of their serene surroundings.

- A colorful scene at the shoreline with a red crab sitting on the golden sand, beside a vibrant turquoise surfboard. The sun, resembling a massive, glowing orange orb, hangs low in the sky, which is painted with a spectrum of a rainbow's hues. Thought bubbles appear above the crab, filled with the words 'you are all that matters', hinting at a whimsical, introspective moment on this picturesque beach.

Figure 4:

- An aerial view of Toronto's skyline dominated by the iconic CN Tower standing tall amongst the surrounding buildings. The image is taken from the window of an airplane, providing a clear, bird's-eye perspective of the urban landscape. Across the image, the words "The CN Tower" are prominently displayed in the playful Comic Sans font. The cluster of city structures is neatly bisected by the glistening blue ribbon of a river.

- A vibrant parrot perched confidently on the wooden railing of an old pirate ship, its feathers a bold mixture of greens, blues, and reds. It's donned a small, comically endearing pirate hat atop its head. The backdrop is filled with the taut ropes and billowing sails of the ship, and emblazoned across the image is a humorous caption declaring, "I'm the captain now."

- An elegant exhibition announcement poster designed in a stylish, contemporary art style, emphasizing clarity and sophisticated textual arrangement. At the top-center, prominently displayed in graceful modern letters, the title states clearly: "Echoes: Modern Art Exhibition". Directly under it, slightly smaller, lies an inviting subtitle in simple serif typography: "Exploring the dialogue between space and emotion through contemporary installations". At the poster's midpoint, concise distinctive textual information arranged horizontally in three segments appears neatly: "Interactive Installations", "Live Artist Talks", and "Exclusive Guided Tours", each paired with minimalistic symbolic illustrations (geometric shapes for installations, speech bubble for artist talks, map pointer icon for tours). Clearly organized at the bottom-third section, precise logistical information reads: "November 14 – December 20, 2023 City Center Art Gallery Open Daily 10 AM - 8 PM". A subtle yet inviting handwritten note placed elegantly at the bottom right encourages visitors: "Experience art beyond boundaries"

- A lively comic-style illustration depicting two humorous cartoon dogs interacting near a freshly dug backyard hole surrounded by scattered dirt, garden tools, blooming flowers, and a wooden fence background. At the upper-left side, Dog One stands nervously near the messy hole, ears down and eyes wide open with an expression of concern. Its speech bubble is an oval shape, outlined neatly with smooth, slightly rounded corners, positioned clearly above Dog One's head. Inside, clearly readable playful handwritten-style text emphasizes the dog's worried tone, saying, "You sure the humans won't notice this giant hole here?". Toward the lower-right side, Dog Two sits calmly and confidently with a cheerful, carefree expression, wagging its tail gently. Its speech bubble is rectangular with softly rounded edges, placed slightly overlapping with Dog One's speech bubble to guide the reader naturally downward diagonally across the frame. Dog Two's friendly, humorous response appears in a whimsical italicized comic font, clearly stating, "Relax! We'll just blame it on the neighbor's cat again!" Each speech bubble creats the playful and engaging backyard scene.

- A vibrant outdoor field with lush green grass and neatly painted boundary lines. Numerous athletic men, donned in brightly colored sports attire, are energetically chasing after a spherical ball under the bright daylight. The background is a soft focus, enhancing the dynamic movement of the players in the foreground. Surrounding the playing area, there are scattered equipment and water bottles, indicating a serious game is in progress.

- The image displays a vibrant array of multicolored flowers and lush green leaves clustered at the lower section. In the bottom right corner, a small, round, terracotta pot peeks into the frame, providing a contrast to the natural elements. The floral bouquet features petals ranging from deep purples to bright yellows, with a variety of leaf shapes and sizes nestled amongst them.

Figure 6:

- A man in a suit is standing in front of the window, looking at the bright moon outside the window. The man is holding a yellowed paper with handwritten words on it: "A lantern moon climbs through the silver night, Unfurling quiet dreams across the sky, Each star a whispered promise wrapped in light, That dawn will bloom, though darkness wanders by." There is a cute cat on the windowsill.

**Baselines.** We compare our method with the following baselines:

- **FLUX.1 Lite** (Daniel Verdú, 2024) is an 8-billion-parameter text-to-image transformer model obtained by distilling the larger FLUX.1-dev model to reduce memory use and speed up inference while generating images from text prompts.

- **TinyFusion** (Fang et al., 2025) is a learnable depth pruning method that removes redundant layers from diffusion transformer models via end-to-end learning to produce a shallower model with high recoverability after fine-tuning.

- **HierarchicalPrune** (Kwon et al., 2025) is a position-aware compression framework for large text-to-image diffusion models that hierarchically prunes less essential later model blocks based on their position, preserves key early block weights, and uses sensitivity-guided distillation to transfer knowledge during pruning.

- **PPCL** (Ma et al., 2025) refers to Pluggable Pruning with Contiguous Layer Distillation, a structured pruning framework for diffusion transformer models that identifies and removes contiguous redundant layer intervals using linear probing and similarity analysis, then applies a teacher-student alternating distillation scheme integrating depth-wise and width-wise pruning in a single training phase.

- **Dense2MoE** (Zheng et al., 2025b) is a framework that restructures dense diffusion transformer models into a Mixture-of-Experts (MoE) architecture by replacing feed-forward networks with MoE layers and introducing block-level sparsity, using a multi-step distillation process to reduce activated parameters while preserving model capacity for text-to-image generation.

- **DyDiT** (Zhao et al., 2024) is a diffusion transformer architecture that modulates computation dynamically across diffusion timesteps and spatial tokens by adapting model width and selectively processing tokens based on their complexity to reduce redundant computation during image generation.

We notice that most methods do not release official implementations for FLUX and Qwen-Image, with the exception of FLUX.1 Lite and PPCL. Consequently, we report the results either directly from the original papers or from the re-implementations provided in the PPCL paper (Ma et al., 2025).

**Evaluation Metrics** We compare our method with previous methods on DPG-Bench (Hu et al., 2024), GenEval (Ghosh et al., 2023) and T2I-Compbench (Huang et al., 2025).

- **DPG-Bench** is a benchmark and evaluation suite for text-to-image and multimodal generative models that measures how well models follow dense, complex prompts (i.e., long text descriptions with multiple objects, attributes, spatial relations, and semantic constraints) by providing a set of standardized dense prompts and an automatic evaluation protocol to quantify semantic alignment and generation performance. It is used to compare models' abilities to understand and generate images that match these intricate, high-detail prompt descriptions.

- **GenEval** is an object-focused evaluation benchmark for text-to-image generative models that quantifies how well a model's outputs align with the compositional constraints in prompts by systematically testing and scoring tasks such as object presence, co-occurrence, counting, color attributes, and spatial relationships using a structured set of prompts and an automated detection/verification pipeline.

- **T2I-CompBench** is a compositional text-to-image generation benchmark that provides a large set of structured text prompts designed to test how well text-to-image models translate linguistic constraints about objects, attributes, relationships, and complex scene descriptions into corresponding images, using specialized metric protocols (e.g., detector-based and attribute checks) to measure alignment between the generated images and the intended compositional semantics.

## B. More Visualization Results.

Here, we provide more visualization results on image generation (Figure 8) and 3D asset generation (Figure 9).

## C. Implementation Details.

The implementation and training on Qwen-Image is based on DiffSynth-Studio (Team, 2024). We have released the code in the GitHub repo.

## D. Limitation and Future Works

Although E-DiT achieves competitive performance on various benchmarks across different modalities, its training process on Qwen-Image can still exhibit instability, and the resulting performance is slightly below the baseline on some evaluation metrics. In future work, we plan to further improve training stability and explore other strategies, such as post-training (**??**)

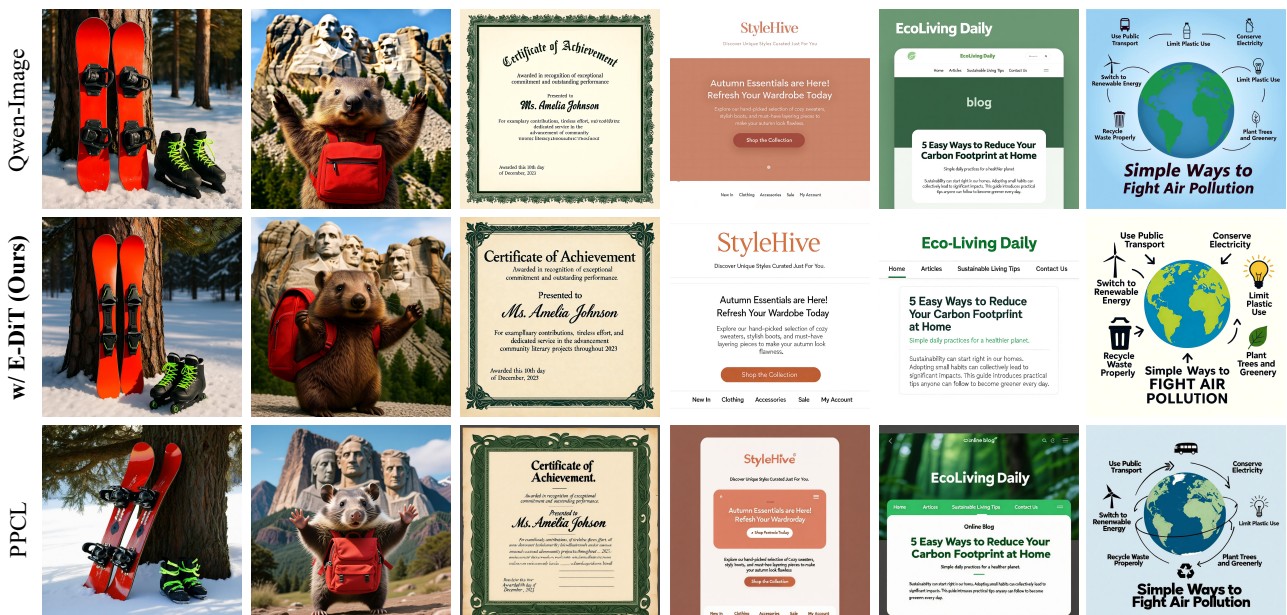

*Figure 8.* More visual comparisons between E-DiT-turbo and open-source baselines.

and CFG-enchancement (Karras et al., 2024; Chen et al., 2025a), to enhance the performance of the accelerated models. The application of E-DiT to various downstream tasks, such as customized generation and editing (Zhu et al., 2026; 2025; Cao et al., 2023; Zhu et al., 2024), is also a promising direction for future research.

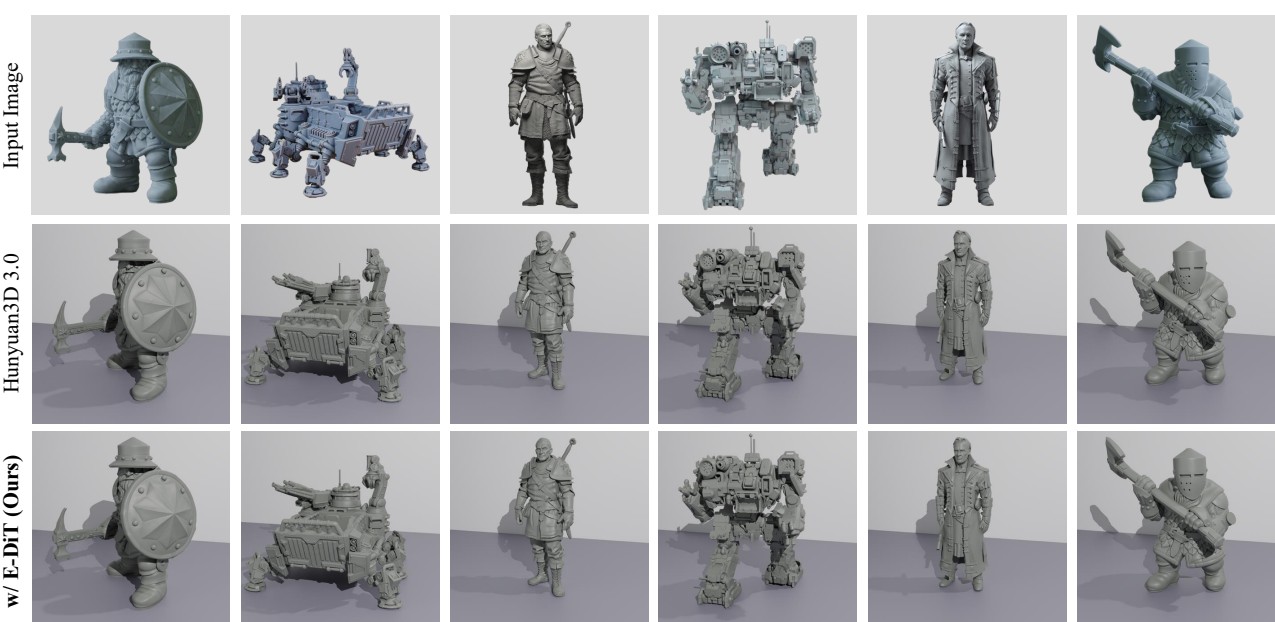

*Figure 9.* More visual comparisons between E-DiT and Hunyuan3d 3.0.

