# OpenReview forum: "Elastic Diffusion Transformer"
_ICML.cc/2026/Conference — ICML 2026 regular_

### Official Review · Reviewer_SJP6 · 2026-03-03

**Soundness:** 3
**Presentation:** 3
**Significance:** 3
**Originality:** 2
**Overall Recommendation:** 4
**Confidence:** 4

**Summary:**

This paper proposes E-DiT, an adaptive acceleration approach (**not training-free**) for DiT, built on the intuitive idea that not every denoising step requires the same amount of computation and that the importance of each block can vary across samples. To address this, the authors insert a lightweight router into every DiT block to decide whether the block can be skipped or, if kept, how much to shrink the MLP width. They also include a block-wise caching mechanism to reuse features for cases where skipping is uncertain. Experiments on Qwen-Image, FLUX.1-dev, and Hunyuan3D-3.0 show roughly 2× inference speedup, with the authors claiming only minor quality degradation.

**Compliance With Llm Reviewing Policy:**

Affirmed.

**Final Justification:**

This paper proposes E-DiT, which speeds up diffusion transformers by adaptively skipping or shrinking blocks using a small router and caching. The idea makes sense and is explained clearly. Experiments show about 2× speedup, which is solid. I also like how they show block importance and timestep sensitivity, it makes the adaptive approach convincing.

I was worried about how new this really is compared to previous router-based methods, missing common metrics, e.g., FID, IS, CLIP-S, and also the training cost and wider applicability. In the rebuttal, the authors have adequately addressed these concerns by clarifying distinctions from prior work and providing additional experiments, which improve confidence in both the contribution and experimental support.

Overall, the paper is solid and clear and seems promising. Even though the originality is a bit limited, I keep it as a weak accept.

**Key Questions For Authors:**

Please see the weaknesses section above.

**Limitations:**

yes

**Strengths And Weaknesses:**

strengths
- The paper is clearly written and well organized. The problem formulation is precise, and the method is described clearly, making the contributions easy to follow.
- Good motivation, with Fig. 2 showing block importance, timestep sensitivity, and compute all varying, making the adaptive approach convincing.
- The authors introduce a block-level router $\mathbf{R}^i$ that smartly predicts which blocks to skip ("MoB") and how much to shrink MLP widths ("MoE"), while reusing features via caching.

weaknesses
- The approach is claimed to be elastic or adaptive, but compared to router-based DyDiT[1] (DyDiT++[2]) and MoB[3], it offers no clear evidence of any real innovation over existing methods.
- The authors claim negligible quality degradation but do not report widely used metrics, e.g., FID, IS, CLIP-S.
- The method requires training, but authors don't discuss the computational cost.
- Applicability to text-to-video models is not discussed.
- Table 4 and Table 5 are incorrectly referenced.

[1]. Wangbo Zhao, Yizeng Han, Jiasheng Tang, Kai Wang, Yibing Song, Gao Huang, Fan Wang, Yang You, Dynamic Diffusion Transformer, ICLR'25;
[2]. Wangbo Zhao, Yizeng Han, Jiasheng Tang, Kai Wang, Hao Luo, Yibing Song, Gao Huang, Fan Wang, Yang You, DyDiT++: Diffusion Transformers with Timestep and Spatial Dynamics for Efficient Visual Generation, TPAMI'26;
[3] Xiaokai Chen, Yongjia Ma, Donglin Di, Lei Fan, Hao Li, Chen Wei, Tonghua Su, ICLR'26 (Submission);

---

> ### Author Rebuttal · Authors · 2026-03-31
>
> Dear Reviewer SJP6,
>
> Thanks for your comprehensive review and insightful comments on our paper! We appreciate that you recognize the clear motivation and impressive results of our method. The response to your concerns is shown below.
> > The approach is claimed to be elastic or adaptive, but compared to router-based DyDiT(DyDiT++) and MoB, it offers no clear evidence of any real innovation over existing methods.
>
> We sincerely thank you for the suggestion, while we believe E-DiT introduces fundamental architectural and methodological advancements that distinguish it from these prior works:
>
> (1). E-DiT and DyDiT accelerate the model with different focuses. Unlike DyDiT, which primarily focuses on attention acceleration and token pruning, E-DiT explores fine-grained block-level acceleration and MLP width pruning. What's more, our E-DiT is orthogonal to DyDiT’s approach; they can be combined, which may achieve even greater efficiency.
>
> (2) MoB predicts skipped blocks solely according to the input prompt. This approach ignores temporal variations across different denoising timesteps and the variations of input latent for each block, which would result in sub-optimal routing. Furthermore, MoB does not explore acceleration of the MLP. In contrast, E-DiT makes local decisions based on the input latent of each specific block, effectively capturing the evolving, fine-grained dynamics of the denoising process. This enables more effective acceleration of both the entire DiT block and the MLP within the block.
>
> (3). MoB relies on a distillation-based framework requiring a pre-trained teacher model, which significantly increases the training overhead. E-DiT eliminates the need for a teacher model, offering a more streamlined and resource-efficient training pipeline.
>
>
>
>
> > The authors claim negligible quality degradation but do not report widely used metrics.
>
> We appreciate the reviewer’s suggestion to include standard quantitative evaluations. We have evaluated E-DiT on the MS-COCO validation set using FID and CLIP score.
>
> As shown in Table 12, E-DiT maintains competitive performance compared to the base model. These results illustrate that our method achieves notable acceleration without sacrificing the generative quality or semantic alignment of the original model. We will also add these results to the revised version of our paper.
>
> **Table 12.** Comparison with baselines on more metrics.
> |Method|Speedup|FID|CLIP|
> |-|-|-|-|
> |FLUX|1x|23.98|31.09|
> |E-DiT (Ours)|2x|23.79|31.04|
>
> > The method requires training, but authors don't discuss the computational cost.
>
> We thank the reviewer for pointing this out. For the Qwen-Image experiment, E-DiT was trained for 8,000 steps on 32 H20 GPUs (batch size of 1 per GPU), totaling approximately 12 hours. This computational cost is comparable to the existing fine-tuning-based methods such as [1,2,3]. Importantly, unlike prior works that often require complex multi-stage procedures, E-DiT supports a fully end-to-end training pipeline, significantly simplifying implementation while maintaining state-of-the-art efficiency. We will add detailed information in the Experiment Setup section to the revised manuscript.
>
>
> > Applicability to text-to-video models is not discussed.
>
>
> Thanks for your insightful advice! To demonstrate the versatility of E-DiT, we have conducted additional experiments on the Wan2.1-1.3B video generation model. The results in Table 13 illustrate that E-DiT generalizes well to the video domain, achieving a $2\times$ acceleration with negligible degradation in temporal consistency or per-frame quality. This successful extension demonstrates that the principles of E-DiT are robust across different modalities and model architectures.
>
> **Table 13.** Results of E-DiT on Wan2.1-1.3B. Experiments are conducted on 480p, 49-frame video generation.
> ||Latency (ms)|VBench|
> |-|-|-|
> |Wan2.1-1.3B|3507|0.813|
> |+E-DiT (Ours)|1853|0.809|
>
> > Table 4 and Table 5 are incorrectly referenced.
>
> We sincerely thank you for your careful reading! We will correct this in the revised version of our paper.
>
>
> References:
>
> [1]. Dense2moe: Restructuring diffusion transformer to moe for efficient text-to-image generation, ICCV 2025.
>
> [2]. DyDiT++: Diffusion Transformers with Timestep and Spatial Dynamics for Efficient Visual Generation, TPAMI.
>
> [3]. Pluggable Pruning with Contiguous Layer Distillation for Diffusion Transformers, CVPR 2026.

---

> > ### Author Rebuttal · Reviewer_SJP6 · 2026-04-02
> >
> > Thank you for the response and additional experiments! I will maintain the score.

---

### Official Review · Reviewer_8xDb · 2026-03-07

**Soundness:** 3
**Presentation:** 3
**Significance:** 2
**Originality:** 3
**Overall Recommendation:** 4
**Confidence:** 3

**Summary:**

This paper proposes the Elastic Diffusion Transformer (E-DiT), an adaptive acceleration framework for Diffusion Transformers (DiTs). The framework equips each Transformer block with a lightweight router to enable: (i) dynamic skipping of blocks, and (ii) adaptive reduction of the MLP width in each block based on the latent representation and timestep. Additionally, E-DiT introduces a training-free, block-level caching mechanism that leverages the router’s predictions to reuse residual features of blocks identified as "marginally important" across adjacent timesteps. Experiments on image generation (Qwen-Image, FLUX) and 3D asset generation (Hunyuan3D-3.0) demonstrate that E-DiT achieves up to approximately 2× inference acceleration with negligible or minimal quality degradation. Ablation studies further validate the effectiveness and individual contributions of each component.

**Compliance With Llm Reviewing Policy:**

Affirmed.

**Final Justification:**

The authors answered all my questions, and I have no further questions, but I suggest that the authors further explain the limitations of the method.

**Key Questions For Authors:**

1. Despite the caching mechanism being a core contribution, the paper lacks direct comparisons with existing caching baseline methods.
2. Latency comparisons for certain baselines appear sourced from external papers or re-implementations (e.g., PPCL benchmark), potentially conducted on differing hardware/software stacks; thus, the fairness of wall-clock time comparisons is questionable.
3. Although the caching hyperparameter δ is shown to be critical, the paper offers limited guidance on its configuration and insufficient robustness analysis across models or datasets.
4. Base models were seemingly not fine-tuned under the same protocol (100,000 images) used for E-DiT router training; reported quality improvements may therefore partially stem from additional fine-tuning rather than solely from E-DiT, introducing a confounding factor.

**Limitations:**

Lacking a limitation discussion.

**Strengths And Weaknesses:**

1. Introduces a unified, sample-adaptive computation framework for DiTs, guided by a lightweight router, jointly enabling dynamic block skipping and per-block MLP width selection.
2. Proposes a simple yet effective router-informed block-level feature caching criterion, eliminating the need for hand-crafted heuristics commonly used in training-free caching methods.
3. Evaluated across multiple strong backbone models—including Qwen-Image and FLUX—and extended to cross-modal tasks such as image-to-3D generation using Hunyuan3D-3.0, strongly supporting the claim of broad applicability.

---

> ### Author Rebuttal · Authors · 2026-03-31
>
> Dear Reviewer 8xDb,
>
> Thank you for your comprehensive and detailed review of our paper and the recognition of our work's clarity and effectiveness! We provide our feedback as follows.
>
> > the paper lacks direct comparisons with existing caching baseline methods.
>
> We sincerely appreciate the reviewer’s constructive suggestion to further improve our work! We have conducted extensive comparative experiments between E-DiT and several state-of-the-art caching baselines, including PreciseCache[1] and TaylorSeer[2].
>
> **Table 10.** Comparison with Cache-based methods on Qwen-Image.
> |Method|Speedup|Geneval|DPG|
> |-|-|-|-|
> |TaylorSeer|1.8x|0.841|84.1|
> |PreciseCache|1.8x|0.839|84.5|
> |E-DiT (Ours)|1.5x|0.893|88.1|
> ||2x|0.853|85.4|
>
> As illustrated in Table 10, E-DiT consistently outperforms these cache-based methods in both computational efficiency and generation quality. We will incorporate these results into the revised manuscript.
>
> > the fairness of wall-clock time comparisons is questionable.
>
> We thank the reviewer for the detailed reviews! Since several baseline implementations are not open-sourced, many previous works such as [3,4] also directly reuse some results reported in the external papers. Therefore, we believe that reporting the results from external papers is reasonable and appropriate. We also conducted a rigorous verification of the reported experimental results to ensure fair comparisons. First of all, we confirmed that the results in external papers reported in our paper were measured on **a single H20 GPU** as mentioned in their papers, which strictly aligns with our own testing setting. Second, we locally measured the inference speed of the base models (e.g., FLUX), ensuring that these results are consistent with those reported in external papers.
>
> As a result, we believe that the comparisons presented in our paper are reliable and fair, effectively demonstrating the effectiveness of our method.
>
> > Although the caching hyperparameter δ is shown to be critical, the paper offers limited guidance on its configuration and insufficient robustness analysis across models or datasets.
>
> We sincerely thank the reviewer for the insightful suggestion. We observe that the effects of different values of δ on the final results are similar across different datasets and models. This is because the architecture and training strategy of the router is consistent across models, resulting in similar routing behaviors (e.g., the magnitude of predicted probabilities) even when the base models differ. In practice, δ is typically set to a value slightly above 0.5 to distinguish blocks whose importance lies near the borderline, without the need for complicated parameter tuning. For different models, setting δ to 0.01 or 0.015 is sufficient to achieve noticeable acceleration without sacrificing quality (as illustrated in Figure 6), and it can be set to a larger value if more aggressive acceleration is desired. We will add more discussions in Section 4.4 in the revised version of our paper.
>
> > Base models were seemingly not fine-tuned under the same protocol (100,000 images) used for E-DiT router training; reported quality improvements may therefore partially stem from additional fine-tuning rather than solely from E-DiT, introducing a confounding factor.
>
> We are grateful for this thoughtful observation. We would like to point out that it is standard practice in most previous works such as [3,4,5] to compare the accelerated models directly against the original base models. We also provide the comparison in Table 11 between a fully fine-tuned base model and E-DiT. The results illustrate that our E-DiT still achieves competitive performance compared with the finetuned baselines.
>
>
> **Table 11.** Comparison with finetuned baselines.
> |Method|Speedup|Geneval|DPG|
> |-|-|-|-|
> |Qwen-Image|1x|0.870|88.9|
> |Qwen-Image-finetuned|1x|0.886|88.8|
> |+E-DiT (Ours)|1.5x|0.893|88.1|
> ||2x|0.853|85.4|
>
> > Lacking a limitation discussion.
>
> We sincerely thank you for your advice. Following your suggestion, we will add a section discussing the limitations of our method in the revised version of the paper.
>
>
> References:
>
> [1]. PreciseCache: Precise Feature Caching for Efficient and High-fidelity Video Generation, ICLR 2026.
>
> [2]. From reusing to forecasting: Accelerating diffusion models with taylorseers, ICCV 2025.
>
> [3]. Dense2moe: Restructuring diffusion transformer to moe for efficient text-to-image generation, ICCV 2025.
>
> [4]. Pluggable Pruning with Contiguous Layer Distillation for Diffusion Transformers, CVPR 2026.
>
> [5]. DyDiT++: Diffusion Transformers with Timestep and Spatial Dynamics for Efficient Visual Generation, TPAMI.

---

> > ### Author Rebuttal · Reviewer_8xDb · 2026-04-02
> >
> > Thank you for the reply. I have no further questions. I suggest the author further explain the limitations of this article.

---

### Official Review · Reviewer_vptR · 2026-03-12

**Soundness:** 3
**Presentation:** 3
**Significance:** 2
**Originality:** 3
**Overall Recommendation:** 4
**Confidence:** 4

**Summary:**

This paper proposes to accelerate diffusion transformers by adaptive adjust the model block and width. A router mechanism is designed to determine the blocks to be skipped, as well as the MLP width of the remaining active blocks. Furthermore, a block-wise caching mechanism is introduced to reuse features across timesteps. The experimental results demonstrate that the proposed method accelerates the inference process of diffusion transformers.

**Compliance With Llm Reviewing Policy:**

Affirmed.

**Final Justification:**

I thank the authors for the detailed clarifications and the extra experimental validation. This resolves my previous questions, and I maintain my positive score unchanged.

**Key Questions For Authors:**

1. Could the paper details the exact consumption of training resources of the proposed method and provide a comparison with the fine-tuning-based baselines?

2. Could the paper provides a trade-off curve demonstrating how the model's generation quality and latency change with continuous variations in the target budget hyper-parameters $\rho_g$ and $\rho_w$ ?

3. Could the paper provides more details of the experimental results during inference? For instance, what is the average number of activated blocks on different benchmarks, and what is the average block width across different layers? Providing these statistical details would be beneficial for readers to understand the learned routing behaviors.

**Limitations:**

Yes

**Strengths And Weaknesses:**

Strengths:
1. The paper is well-written and the methodology is clearly presented.

2. Comprehensive experiments are conducted across multiple tasks and DiT models.

Weaknesses:
1.  The proposed method relies on predefined target budgets, specifically $\rho_g$ and $\rho_w$ for block skipping and for MLP width reduction. While the authors provide specific configurations, the paper lacks a systematic analysis of how the model's performance smoothly changes as these budget parameters vary.

2. The proposed method requires an end-to-end fine-tuning phase. Although the dataset used is relatively small, the experiments are conducted on a 32-GPU cluster. The paper needs to report the training resources consumed by the proposed method and compare it with other baselines to prove its training efficiency.

---

> ### Author Rebuttal · Authors · 2026-03-31
>
> Dear Reviewer vptR,
>
> Thanks for your time and thoughtful review! We sincerely appreciate your recognition of the experimental performance and clear writing of the paper. We provide our feedback as follows.
>
> > Could the paper details the exact consumption of training resources of the proposed method and provide a comparison with the fine-tuning-based baselines?
>
> Thank you very much for this helpful suggestion! For Qwen-Image, E-DiT is trained for 8,000 steps on 32 H20 GPUs, with a per-GPU batch size of 1, requiring approximately 12 hours in total. This resource consumption is comparable to previous fine-tuning–based methods such as [1,2,3]. Notably, our approach supports **fully end-to-end training**, without the need for multi-stage procedures adopted in many prior works, which significantly simplifies the training pipeline.
>
> > Could the paper provides a trade-off curve demonstrating how the model's generation quality and latency change with continuous variations in the target budget hyperparameters?
>
> Thank you for the insightful suggestion! We have provided ablation results on $ρ_g$ and $ρ_w$ in Table 8 and Table 9. These experiments illustrate that setting these hyperparameters too high would lead to relatively insufficient acceleration, whereas setting them too low may degrade generation quality. We will further add these discussions in the revised manuscript.
>
>
> **Table 8.** Ablation Study on $ρ_g$.
> |$ρ_g$|Latency (ms)|Geneval|
> |-|-|-|
> |Qwen-Image|2431|0.870|
> |0.45|1301|0.782|
> |0.5|1643|0.857|
> |0.6|1874|0.871|
>
> **Table 9.** Ablation Study on $ρ_w$.
> |$ρ_w$|Latency (ms)|Geneval|
> |-|-|-|
> |Qwen-Image|2431|0.870|
> |0.5|1492|0.798|
> |0.6|1643|0.857|
> |0.7|1796|0.861|
>
>
> > Could the paper provides more details of the experimental results during inference? For instance, what is the average number of activated blocks on different benchmarks, and what is the average block width across different layers?
>
> Thanks for the insightful suggestion! We report the average number of activated blocks across benchmarks and the average effective MLP width across layers in Table 9. On average, about 40% of the blocks are skipped, while the remaining 60% operate at approximately 66% of the original MLP width. We will add these details into our paper to improve clarity and completeness.
>
> **Table 9.** More detailed inference analysis.
> |Benchmark|Activated blocks|MLP width|
> |-|-|-|
> |DPG|62%|67%|
> |Geneval|60%|66%|
>
>
> References:
>
> [1]. Dense2moe: Restructuring diffusion transformer to moe for efficient text-to-image generation, ICCV 2025.
>
> [2]. DyDiT++: Diffusion Transformers with Timestep and Spatial Dynamics for Efficient Visual Generation, TPAMI.
>
> [3]. Pluggable Pruning with Contiguous Layer Distillation for Diffusion Transformers, CVPR 2026.

---

> > ### Author Rebuttal · Reviewer_vptR · 2026-04-02
> >
> > I thank the authors for the detailed clarifications and the extra experimental validation. I have no further questions and will retain my score.

---

### Official Review · Reviewer_9q5X · 2026-03-15

**Soundness:** 3
**Presentation:** 3
**Significance:** 3
**Originality:** 3
**Overall Recommendation:** 4
**Confidence:** 4

**Summary:**

This paper introduces the Elastic Diffusion Transformer (E-DiT), a method to accelerate inference in diffusion transformers by dynamically adjusting the computational graph on a per-sample and per-timestep basis. The core contribution is a lightweight router module added to each transformer block, which predicts two things: whether the entire block can be skipped, and what fraction of the MLP's hidden dimension should be used if the block is active. The model is trained end-to-end on top of a pre-trained diffusion backbone using a joint objective that combines the standard diffusion loss with efficiency regularization losses. These regularizers encourage the routers to meet specified block-skipping and MLP-width reduction budgets. During inference, E-DiT further leverages a block-wise caching mechanism that reuses previously computed residuals for blocks with borderline skipping probabilities, exploiting temporal redundancy across denoising steps. The method is evaluated on several state-of-the-art models, including Qwen-Image, FLUX, and Hunyuan3D, demonstrating significant inference speedups (e.g., up to 2.2x) with practically no loss in generation quality across image and 3D asset generation tasks.

**Compliance With Llm Reviewing Policy:**

Affirmed.

**Key Questions For Authors:**

1. Router Ablation: How critical is the router architecture's design? Have you experimented with simpler router designs, such as using only a single linear layer on the pooled timestep+conditioning embedding, or using the [CLS] token from the block's input? What motivated the specific choice of using mean-pooled latent features combined with the timestep embedding?

2. Training Dynamics of the Straight-Through Estimator: The paper uses a straight-through estimator for the gating and width decisions. Did you encounter any challenges with training stability or convergence when using this estimator, especially in the early stages of training when the routers are "fully open"? If so, how did you address them (e.g., learning rate scheduling, gradient clipping)?

3. Overhead Analysis: The paper reports impressive per-step latency improvements. Could you provide a more detailed breakdown of the source of these gains? Specifically, what is the added inference cost of the router itself, and what is the overhead of the block-wise caching mechanism (e.g., checking the cache, storing residuals)? Is the router's cost amortized by the savings from skipping blocks and reducing MLP width?

4. Generalization to Other Modalities: The method is successfully applied to 2D image and 3D asset generation. Do you foresee any challenges in applying it to video generation models (like HunyuanVideo or CogVideoX), which have an additional temporal dimension? Would the router need to be adapted to handle the extra complexity, or could the current design be applied directly to each frame's processing?

**Limitations:**

No. The paper does not include a dedicated "Limitations" or "Broader Impact" section. While the authors mention in the "Impact Statement" that there are no specific consequences they feel need highlighting, a more thorough discussion of limitations would strengthen the paper. For example, it could discuss the potential for the router to make suboptimal decisions for out-of-distribution prompts, the slight increase in model complexity due to the added routers, or the difficulty of tuning the budget parameters for a new model, task or domain. Adding a short paragraph reflecting on these points would demonstrate a balanced and honest assessment of the work.

**Strengths And Weaknesses:**

The proposed method is clearly motivated and well-integrated with existing diffusion transformer architectures (DiT, MMDiT). The training objective, combining a standard diffusion loss with carefully designed efficiency losses (gating and width), is appropriate for the goal. The use of a straight-through estimator to handle the non-differentiable sampling steps is a standard and valid technique. The experiments are well-designed, comparing against a range of relevant baselines, including both static pruning methods (e.g., PPCL, FLUX.1 Lite) and dynamic methods (e.g., DyDiT). The evaluation covers multiple modalities (2D image and 3D asset generation) and uses established benchmarks (DPG-Bench, GenEval, T2I-CompBench, ULIP), lending credibility to the claims.

While the experiments are broad, a deeper analysis of the router's decisions could strengthen the soundness. For instance, visualizing which blocks are skipped or have reduced width across different timesteps for various prompts would provide intuitive evidence that the router is learning meaningful, content-dependent patterns. The reliance on straight-through estimators, while standard, can sometimes lead to unstabie training or suboptimal gradient signals; a brief discussion or ablation on its impact would be beneficial. The comparison to DyDiT, while included, is not a 100% fair comparison as DyDiT is trained from scratch, whereas E-DiT is fine-tuned from a pre-trained model. The paper could acknowledge this difference more explicitly. Furthermore the results are lying within the range of what other pruning and distillation methods have achieved - what architectural insight would the authors possess as the major one?

---

> ### Author Rebuttal · Authors · 2026-03-31
>
> Dear Reviewer 9q5X,
>
> We sincerely thank you for the comprehensive review! We appreciate your recognition of the motivation and performance of E-DiT. Here is our feedback:
>
>
> > Router Ablation: How critical is the router architecture's design?
>
> Thanks for this insightful suggestion! In fact, we notice that the router architecture has a significant impact on the final performance.
>
> Our decision to adopt mean-pooled latent features combined with the timestep embedding is motivated by two considerations. First, average pooling yields a compact yet informative feature vector that efficiently summarizes the entire latent representation. Second, inspired by prior work such as [1,2], we notice that timestep information plays an essential role in determining redundancy, and incorporating the timestep embedding would greatly improve the router’s decision quality. On the other hand, an overly simplistic router architecture, such as a single linear layer, leads to highly unstable training and often fails to converge.
>
> We also provide an ablation on the channel size of our router in Table 6, reporting the inference speed of a single router in Qwen-Image backbone when generating a 1024\*1024 image on a single H20 GPU. The results indicate an overly small channel limits the router’s capacity, degrading routing quality and ultimately harming performance. Some visualizations of the router prediction are also demonstrated in Figure 7.
>
> **Table 6.** Ablation Study of the Router Channel Size.
> |Channel|Speed (ms)|Geneval|
> |-|-|-|
> |32|0.3|0.801|
> |128 (Ours)|0.5|0.853|
> |512|1.3|0.857|
>
>
>
> > Training Dynamics of the Straight-Through Estimator
>
> We sincerely appreciate your thoughtful question! Training stability is a central concern when using the Straight-Through Estimator (STE). Our observations show that initializing the router in a fully open state is critical for stable optimization (as illustrated in Table 5). With this initialization, the STE-based training process is generally stable: early in training, the performance loss remains low due to all blocks being active, while the FLOP loss is initially high but gradually converges toward the target budget. When the router begins skipping blocks and reducing MLP width, there is an increase in performance loss, but this effect diminishes as training progresses and the model converges.
> Conversely, when the router is initialized randomly, the training process becomes unstable and convergence is significantly more difficult. This reinforces the importance of the “fully open” initialization strategy.
>
> > Overhead Analysis
>
> Thank you for pointing out this important aspect! As shown in Table 4, we provide a detailed ablation study and comparisons with static pruning baselines. All three contributions of E\-DiT play essential and complementary roles in achieving the final acceleration.
>
> Regarding overhead, the inference cost of the router itself is reported in Table 6 and is negligible relative to the total inference runtime. Moreover, the block-wise caching mechanism introduces nearly no additional computational cost. This is because the cache lookup and residual storage are extremely lightweight operations in Python and do not bottleneck the pipeline. As a result, the computational cost of all additional modules introduced in E-DiT is so small that it can be neglected during the generation process.
>
> > Generalization to Other Modalities
>
> We appreciate the reviewer’s forward-looking question! E-DiT naturally extends to video generation models, and the router architecture requires no modification to handle the temporal dimension, since video data is also represented as a sequence of tokens in the generation process, similar to image and 3D generation. We conducted experiments on Wan-1.3B, and the results show that E-DiT maintains strong performance in the video domain, achieving nearly 2× acceleration with negligible quality degradation. This suggests that E-DiT generalizes effectively across modalities.
>
> **Table 7.** Results of E-DiT on Wan2.1-1.3B. Experiments are conducted on 480p, 49-frame video generation.
> ||Latency (ms)|VBench|
> |-|-|-|
> |Wan2.1-1.3B|3507|0.813|
> |+E-DiT (Ours)|1853|0.809|
>
> > The comparison to DyDiT, while included, is not a 100% fair comparison, as DyDiT is trained from scratch
>
> Thanks for your advice! According to the paper of DyDiT and DyDiT++, we would like to point out that the performance of DyDiT reported in our paper is also **obtained by finetuning from a pretrained checkpoint**. As a result, we believe the comparison between DyDiT and E-DiT is fair.
>
> > Discussions of Limitations
>
> We sincerely thank you for your advice! Following your suggestion, we will add a section discussing the limitations of our method in the revised version of the paper.
>
>
> References:
>
> [1]. Dynamic diffusion transformer, ICLR 2025
>
> [2]. PreciseCache: Precise Feature Caching for Efficient and High-fidelity Video Generation, ICLR 2026

---

> > ### Author Rebuttal · Reviewer_9q5X · 2026-04-06
> >
> > The authors provided answered to my questions

---

### Decision · Program_Chairs · 2026-04-30

**Decision:**

Accept (regular)

**Comment:**

The paper initially received mixed reviews with 2 Weak Reject and 2 Weak Accept scores. The rebuttal addressed all reviewers' concerns, leading to all final scores as Weak Accept. The paper was well-received for its clear motivation, appropriate solution, good writing, and comprehensive experiments across multiple tasks and DiT models, yielding strong results.

The ACs reviewed and agreed with the reviewers' decision to accept the paper.